# Atomic Insight into the Nano-Grinding Mechanism of Reaction-Bonded Silicon Carbide: Effect of Abrasive Size

**DOI:** 10.3390/mi16091049

**Published:** 2025-09-15

**Authors:** Honglei Mo, Xie Chen, Cui Luo, Xiaojiang Cai

**Affiliations:** Shanghai Aerospace Control Technology Institute, Shanghai 201109, China; mohonglei1987@sina.com (H.M.); spwanx@163.com (X.C.); luocui16@163.com (C.L.)

**Keywords:** molecular dynamics simulation, nano-grinding, reaction-bonded silicon carbide, nanoscale machining mechanism, abrasive size

## Abstract

Reaction-bonded silicon carbide (RB-SiC) is a high-performance ceramic material known for its excellent mechanical, thermal, and chemical properties. It contains phases with different mechanical properties, which introduce complex machining mechanisms. In the present work, molecular dynamics (MD) simulation was conducted to investigate the effect of abrasive size on the nano-grinding mechanism of RB-SiC. The surface morphology and subsurface deformation mechanism were investigated. The simulation results suggest that when a small abrasive is used, the surface swelling of SiC is primarily generated by the bending and tearing of SiC at the interfaces. As the abrasive radius increases, the surface swelling is mainly formed by Si atoms, which is identified as elastic recovery. Meanwhile, the material removal rate gradually decreases, and the depth of plastic deformation is obviously increased. Stocking of Si is more apparent at the interface, and obvious sliding of SiC grains is observed, forming edge cracks at the margin of the workpiece. In the subsurface workpiece, the high-pressure phase transition (HPPT) of Si is promoted, and the squeeze of disordered Si is obvious with more dislocations formed when larger abrasive is used.

## 1. Introduction

Reaction-bonded silicon carbide (RB-SiC) is a high-performance ceramic material known for its excellent mechanical, thermal, and chemical properties. It is widely used in the optical mirrors of micro-electromechanical systems, high-energy laser modules, and photoelectric detectors [1]. To meet the surface accuracy requirements in these applications, nanoscale machining methods, including nano-grinding and nano-polishing, have attracted great research interest in the field of ultra-precision machining (UPM) [2,3]. Different to macroscale machining, the material removal thickness in UPM ranges from tens to hundreds of nanometers, which introduces distinct deformation mechanism of workpiece material [4]. The microstructures in RB-SiC, such as grain/phase boundaries, have a significant influence on the material removal process. Therefore, an in-depth investigation of the deformation mechanism at nanoscale is essential to reveal the physical nature in UPM of RB-SiC.

Due to the difficulty faced in the experimental observation of the material removal process, numerical simulation is widely used in investigations of the machining mechanism in UPM. Owing to the rapid development of computational science, molecular dynamics (MD) simulation has emerged as an effective implement to investigate nanoscale deformation features, including phase transition [5], dislocation [6], and cracks [7]. It has been widely used to explore the deformation mechanism of various materials in UPM [8,9]. For instance, Wang et al. [10] studied the removal behavior of single-crystal Si during nanometric cutting via MD simulation. They revealed that contact-induced amorphization of Si dominates the extrusion process, while strain-induced amorphization is responsible for shear removal. Meng et al. [11] investigated the friction behavior of single-crystal 3C-SiC during nano-scratching using MD simulation. They observed structure transition from order to disorder phases, coupled with dislocations in the subsurface workpiece in nano-cutting. Sahoo et al. [12] employed MD simulation to investigate the scratch damage of silica glass against a rigid diamond indenter. They found that densification and shear flow play a major role in the formation of scratch damage. Bhamra et al. [13] used MD simulation with a reactive force field to investigate the wear of single-crystal diamond tips sliding on α-quartz surfaces covered with water molecules. They found that the diamond wear on α-quartz is caused by the formation of C-O interfacial bonds and C-C cleavage, coupled with the diffusion of carbon into the substrate and oxidation.

For polycrystalline materials, the deformation mechanism can be different from that of single crystals. Pradhan et al. [14] studied the wear-resistance behavior in polycrystalline Pt-Au under a hard grinding ball using MD simulation. They discussed the effect of adding atomic concentrations of Au at the grain boundaries (GBs) on the wear process. Their results indicate that the segregation of Au to the GBs of nanocrystalline Pt leads to an improvement in the slab’s anti-wear behavior, suggesting that the underlying wear mechanism is adhesion-dominated. Goel et al. [15] conducted an MD simulation of nanometric cutting on polycrystal Si and revealed that GBs facilitate the amorphization and formation of irregular nanogrooves on the machined workpiece. Liu et al. [16] investigated the material removal behavior of polycrystal 3C-SiC during nano-scratching using MD simulation. They revealed that grain geometry and GBs in the polycrystalline workpiece could affect the internal stress state and suppress the phase transition process, which causes different subsurface damage patterns with single crystals. Meng et al. [17] built an MD model to study the friction mechanism during the nano-grinding of hot-pressed SiC, which is composed of 3C-, 4H-, and 6H-SiC grains. They revealed that the machinability of hot-pressed SiC is determined by the microstructure-related intergranular slip motion and grain fragmentation. Recently, Liu et al. [18] conducted grooving experiments and MD simulation to study the machining mechanisms of RB-SiC under various temperatures. Their results indicate the significance of property mismatch between phases in nanoscale machining. For brittle materials such as Si and SiC, the material removal mechanism is determined by the tool geometry parameters, including rake angle and edge radius [19,20]. However, in the above-mentioned research, the effect of tool geometry on the material removal behavior of RB-SiC was not mentioned, and the coupling deformation mechanism of Si and SiC phases under different tool shapes has not been fully revealed.

In the present work, MD simulation was conducted to explore the nano-grinding mechanism of RB-SiC. The effect of abrasive size on the deformation behavior at the phase boundaries was discussed in detail. A visual image of the material removal process and subsurface damage evolution during the nano-grinding process are established with consideration of the property mismatch between phases. This study contributes to improving our understanding in the machining mechanism of RB-SiC during UPM and provides insight into the coupling deformation behavior of different phases in nanoscale/macroscale machining.

## 2. Methodology

Figure 1 shows the adopted nano-grinding model, which contains a deformable workpiece and a rigid diamond abrasive. The polycrystal structure of the workpiece was constructed using the Voronoi algorithm from Atomsk [21,22]. The same amounts of Si and SiC grains were set in the workpiece, with random distributions. The average size of the SiC grains is 15.2 nm. Following the classic setup in the MD simulation of UPM [23], the workpiece atoms were classified into three groups: the fixed group, the thermostat group, and the Newtonian group. A periodic boundary condition was applied in the *y* direction, while a fixed boundary condition was used in the *x* and *z* directions. The analytical bond order potential (ABOP) [24] was used to describe the atomic interaction in the simulation systems. It is widely used in simulations of UPM on Si and SiC [25,26] since it provides a desirable description of the mechanical properties of diamond and diamond-like structures [27,28]. The energy function of ABOP is described as follows:
(1)$$U\;=\;\sum_{i\;>\;j}\;f_{C}\;(r_{\mathrm{ij}})[V_{R}(r_{\mathrm{ij}})\;-\;\frac{b_{\mathrm{ij}}\;+\;b_{\mathrm{ji}}}{2}\;V_{A}(r_{\mathrm{ij}})]$$
(2)$$V_{R}(r)=\frac{D_{0}}{S\;-\;1}\exp\;[-\beta \sqrt{2S}(r\;-\;r_{0})]$$
(3)$$V_{A}(r)=\frac{{\mathrm{SD}}_{0}}{S-1}\exp[-\beta \sqrt{2/S}(r\;-\;r_{0})]$$
(4)$$f_{C}(r)=\left\{\begin{array}{cc} \;\;\;\;1 & \;\;\;r\;<\;R-D\\ \frac{1}{2}\;-\;\frac{1}{2}\sin\left(\frac{\pi }{2}\frac{r-R}{D}\right) & \;\;\;\;|\;R\;-\;r\;|\;\leq \;D\\ \;\;\;\;\;\;0 & \;\;\;R+D\;<\;r\; \end{array}\right.$$
*b_ij_* = (1 + *χ_ij_*)^−1/2^(5)
*χ*_*ij*_ = ∑_*k* (≠*i*,*j*)_
*f*_*C*_(*r*_*ik*_)exp[2*μ*(*r*_*ij*_ − *r*_*ik*_)]*g*(*θ*_*ijk*_)(6)
(7)$$\;g(\theta )=\;\gamma [1+\frac{c^{2}}{d^{2}}-\frac{c^{2}}{\;\;d^{2}+{\left(\cos\theta \;+h\right)}^{2}}]$$
where *D*_0_ and *r*_0_ represent the dimer energy and the bond length, respectively; *S* and *β* are determined based on the Pauling plot slope and ground-state oscillation frequency of the dimer; *R* and *D* specify the position and the width of the cutoff region, respectively; and variables *h*, *γ*, *c*, *d*, and 2*μ* are adjusted to describe the three-body interactions. Details of these parameters are presented in Table 1. The Large-scale Atomic/Molecular Massively Parallel Simulator (LAMMPS) [29] was used to perform the simulation. And the Open Visualization Tool (OVITO) [30] was applied to visualize and analyze the results. Detailed information of the simulation is present in Table 2.

## 3. Results and Discussion

### 3.1. Surface Morphology

The height distribution and morphology of the machined workpiece are shown in Figure 2. The property mismatch between Si and SiC introduces obvious surface structures at the phase boundaries, which is distinct to the machined surface on single crystals [31]. Chips and side flow show an apparent irregular profile since different removal mechanisms are involved when machining Si and SiC phases. When small abrasives (*R* = 5 nm and 6 nm) are used, the surface swelling of SiC is observed as the abrasive moves from the SiC to the Si phase due to the bending and tearing of SiC at the interfaces. SiC nanocrystals are observed in chips because of the polycrystallization process. These hard nanocrystals are responsible for the formation of scratch patterns on the Si machined surface. As the abrasive radius increases, no obvious tearing of materials are observed at the interfaces, and the side flow is distributed more uniformly near the machined surface. Scratch patterns are still obvious, indicating the existence of SiC polycrystallization, while the surface swelling is mainly formed by Si atoms, which can be identified as the elastic recovery effect. It is revealed that for brittle materials such as Si and SiC, the highly compressed materials could be restored to their balanced position during unloading, causing deviation of the machined surface to the theoretical value [32].

The cross-sectional view of workpiece is shown in Figure 3, with the Si and SiC phases identified. In the MD simulation, the material removal thickness ranges from several to tens of nanometers, which indicates that the material is probably removed in the ductile mode. It can be concluded that the surface error is mainly caused by elastic recovery and scratch patterns due to the property mismatch between phases. When machining from SiC to Si, the torn SiC generated at the interface could move with the abrasive and collapse by the following SiC grain, while elastic recovery is more apparent when machining from the Si to SiC phase. When abrasives of a larger size are used, cracks are observed at the margin of the workpiece, which is caused by sliding of the SiC grains. To measure the effect of different swelling mechanisms on surface integrity, the surface roughness was calculated based on the positions of the surface atoms, which are identified according to the atomic volume (shown in Figure 4a). The surface atoms usually have a large atomic volume based on the Voronoi algorithm [33]. The surface roughness, *Sa*, can be estimated via the following equation [33]:(8)$$\;\mathrm{Sa}\;=\;\frac{1}{A}\iint \left|z(x,y)\right|dxdy\;\approx \;\frac{1}{n}\sum \left|d\right|$$
where *d* represents the distance between surface atoms and the theoretical surface. Figure 4b shows the calculated roughness of the ground surface with abrasives of different sizes. The surface roughness shows a trend of increasing as the abrasive size increases, indicating the significance of elastic recovery on the overall surface roughness.

Figure 5 shows the calculated material removal rate as the grinding distance reaches 65 nm, which is calculated by counting the number of atoms in the deformation region. It is observed that compared to SiC, the Si phase has a higher removal rate due to lower hardness and brittleness. Meanwhile, the downward sliding of the SiC grains also contributes to the smaller material removal rate of SiC. As the abrasive size increases, the material removal rate gradually decreases. This variation can be explained by the fact that the sharp edges from smaller abrasives suppress the elastic deformation of the workpiece material and promote the concentrated plastic deformation in the machining region.

### 3.2. Deformation Behavior

During nano-grinding, the deformation of RB-SiC is more complex than that of single crystals due to the random distribution of grains and the property mismatch between phases. Figure 6 shows the cross-sectional views of the magnitude distribution of atomic displacement in workpieces, referring to the initial stage. Elastic deformation can be identified as the region where the displacement magnitude grows gradually, while plastic deformation is recognized as the clear interruption of the displacement magnitude since destruction of the crystal lattice occurs. When the abrasive cuts into the grains, the deformation mechanism is similar to that of single crystals. While as the abrasive cuts through the phase boundaries, the residual plastic deformation pattern on the machined surface is non-uniform due to the difference in deformation behavior between phases. It is observed that when machining from Si to SiC, the high-flowability disordered Si atoms are covered on the Si surface and stocked at the interfaces, while due to the insufficient support of Si, obvious bending and plastic deformation of the hard SiC occur when machining from SiC to Si. As the abrasive size increases, the depth of the plastic deformation is obviously increased. The stocking of Si is more apparent at the interfaces, which indicates stronger generation of the disordered phases. For the SiC phase, more dislocation patterns are observed inside the grains as a result of enhanced plastic deformation, and obvious grain sliding is observed, forming edge cracks at the margin of the workpiece.

For a clear view of the crystal structure in the subsurface workpiece, the workpiece structure was identified via improved common neighbor analysis (CNA) [34], as shown in Figure 7. The white atoms are the disordered atoms, which mainly contain amorphous phase and defective structures. It can be seen that SiC nanocrystals are formed on the machined surface and chips due to the polycrystallization process, while most disordered phases are Si atoms generated by the high-pressure phase transition (HPPT). As the abrasive advances, the disordered Si atoms can stock and squeeze into the boundaries at the interfaces. When abrasives with larger size are used, the HPPT of Si is promoted, and the squeeze of the disordered Si becomes more obvious. Figure 8 shows the number of disordered atoms after grinding. It is observed that as the abrasive radius increases, more disordered Si atoms are generated, while the variation of disordered SiC atoms is slight. This result indicates that raising the abrasive radius enhances the HPPT of Si, while its influence on SiC is inapparent.

### 3.3. Dislocation Analysis

From Figure 7, obvious dislocation patterns are observed in the subsurface workpiece. To give a clear view of the dislocation propagation during the nano-grinding of RB-SiC, the dislocation structures of Si and SiC atoms are calculated individually, as shown in Figure 9. Atoms near the grain boundaries are deleted before calculation to eliminate the influence of the initial dislocation on the results. In Si grains, the dominant dislocation styles are perfect dislocation with a Burger’s vector of 1/2<110>, and partial dislocation with a Burger’s vector of 1/6<112>. It can be seen in Figure 9a that several dislocations are formed in the deformed region and the deep subsurface workpiece, which can be attributed to the compression from the deformed SiC grains; while for SiC grains, dislocations with Burger’s vectors of 1/3<1-210> and 1/3<1-100> are observed in the subsurface workpiece as a result of polycrsytallization. The length of the dislocation is counted and is shown in Figure 10. It is clear that more dislocations are formed in the subsurface workpiece as the abrasive size increases, which is attributed to the larger deformed region induced by raising the abrasive size.

### 3.4. Internal Stress

Internal stress is intimately connected to the structural evolution in the subsurface workpiece during UPM. To further investigate the impact of abrasive size on subsurface damage during nano-grinding, the hydrostatic stress and von Mises stress of workpiece atoms are calculated using the following stress tensors [35]:(9)$$\sigma_{\mathrm{hydrostatic}\;}=\;\frac{1}{3}{(\sigma }_{\mathrm{xx}}+\;\sigma_{\mathrm{yy}}+\;\sigma_{\mathrm{zz}})$$(10)$$\sigma_{\mathrm{von}\mathrm{Mises}}=\sqrt{\frac{1}{2}({\left(\sigma_{\mathrm{xx}}\;-\;\sigma_{\mathrm{yy}}\right)}^{2}+{\left(\sigma_{\mathrm{yy}}-\sigma_{\mathrm{zz}}\right)}^{2}+{\left(\sigma_{\mathrm{zz}}-\sigma_{\mathrm{xx}}\right)}^{2}+6(\tau_{\mathrm{xy}}^{2}+\tau_{\mathrm{yz}}^{2}+\tau_{\mathrm{zx}}^{2}))}$$
where *σ_xx_*, *σ_yy_*, *σ_zz_*, *τ_xy_*, *τ_xz_*, and *τ_yz_* are stress tensors. In this simulation, a cubic box spatial average of 2 nm is used to average the stress tensor. Figure 11 shows the stress distribution when machining across different phases. It can be seen in Figure 11a that the compressive stress increases obviously when the abrasive contacts with the hard SiC grains. Meanwhile, the bending of SiC grains causes an obvious increase in tensile stress inside the SiC grains and compressive stress at the interfaces. The increase in von Mises stress also indicates the bending of SiC grains. When the abrasive machines from the Si to the SiC phase (shown in Figure 11b), the compressive region is mainly restrained in the Si phase due to compression from the SiC and diamond abrasive. The slight increase in von Mises stress in SiC is caused by the enhanced extrusion of the disordered Si atoms. When the abrasive size increases, the compression of the workpiece material is significantly enhanced, causing more HPPT. When machining from SiC to Si phases, the bending of SiC becomes more apparent as an obvious increase in von Mises stress is observed in the SiC grains, as shown in Figure 11c. When machining from Si to SiC, the apparent increase in von Mises stress in SiC suggests that the extrusion of disordered Si atoms is enhanced (shown in Figure 11d), which introduces stronger upward atomic flow.

To investigate the dynamic stress state, the average stress in workpiece during grinding was calculated, since varying the abrasive radius could change the scale of the deformation region. A slice with 1 nm thickness in the middle of the workpiece was selected to calculate the average stress, as shown in Figure 12. The calculated average stress is shown in Figure 13. The internal stress increases obviously when the abrasive contacts with the hard SiC grains. When the abrasive size increases, the internal stress raises obviously, especially when machining SiC grains. As the abrasive size increases from 5 nm to 10 nm, the average compressive stress increases by 184%, and the average von Mises stress increases by 140%.

## 4. Conclusions

In the current work, the effect of abrasive size on the machining mechanism of RB-SiC during nano-grinding was investigated using MD simulation. The surface morphology and subsurface deformation mechanism were investigated. Although the simulation scale is much smaller than that in experiments, the results of this simulation provide an atomic insight into the couple defamation behavior of SiC and Si phases, which is important for surface integrity during nanoscale/macroscale machining of RB-SiC. The following is a summary of the main conclusions:

When small abrasives are used, surface swelling of SiC is observed due to the bending and tearing of SiC at the interfaces. As the abrasive radius increases, the surface swelling is mainly formed by Si atoms, which is mainly identified as elastic recovery. The surface roughness shows an increasing trend as the abrasive size increases, indicating the significance of elastic recovery on the overall surface roughness.During nano-grinding, Si has a higher material removal rate than SiC. As the abrasive size increases, the material removal rate gradually decreases, and the depth of plastic deformation is obviously increased. Stocking of Si is more apparent at the interface, and more plastic deformation patterns are observed inside the SiC grains. Furthermore, an obvious sliding of the grains is observed, forming edge cracks at the margin of the workpiece.In the subsurface workpiece, the disordered Si atoms can be stocked and squeezed into the phase boundaries. When large abrasives are used, the HPPT of Si is promoted, and the squeeze of the disordered Si becomes more obvious. More disordered Si atoms are generated, while variation in the disordered SiC atoms is slight.As the abrasive size increases, compression of the workpiece material is significantly enhanced, and more dislocations are formed in the subsurface of the workpiece. The bending of SiC becomes more apparent with an obvious increase in von Mises stress when machining from SiC to Si. Meanwhile, the von Mises stress can be increased by the enhanced extrusion of Si as the abrasive moves from Si to SiC.

## Figures and Tables

**Figure 1 micromachines-16-01049-f001:**
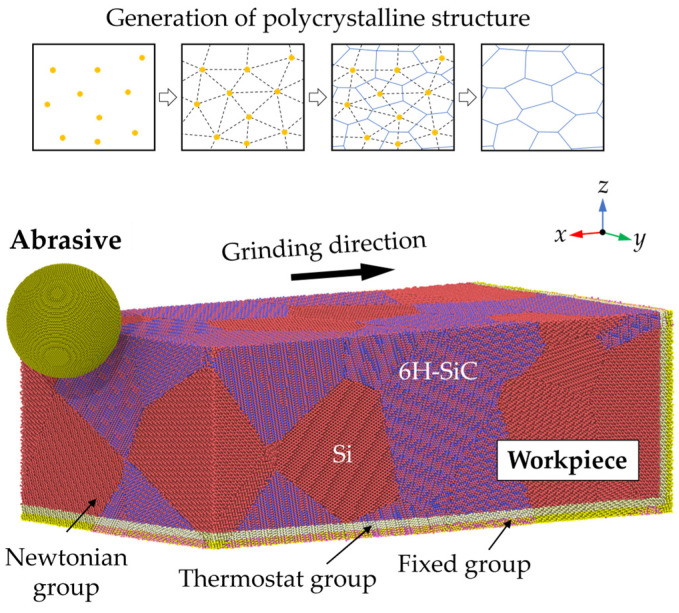
The adopted MD model for nano-grinding simulation.

**Figure 2 micromachines-16-01049-f002:**
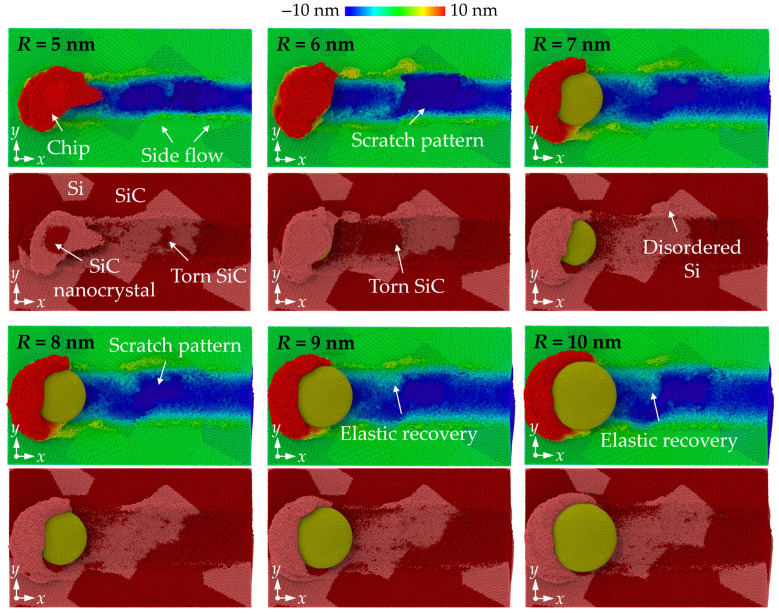
Height distribution and morphology of the ground surfaces machined by abrasives of different sizes.

**Figure 3 micromachines-16-01049-f003:**
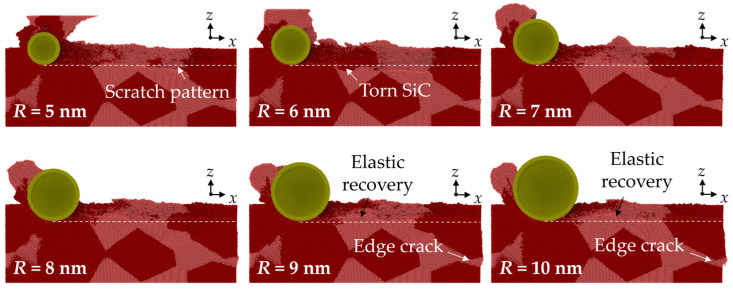
Cross-sectional view of the workpiece machined by abrasives of different sizes.

**Figure 4 micromachines-16-01049-f004:**
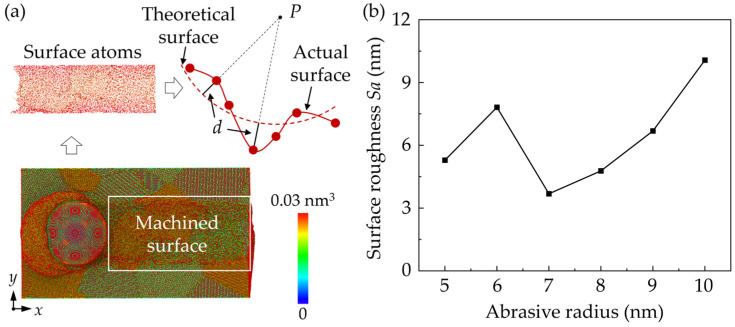
Surface roughness of the ground surface: (**a**) illustration of the surface roughness calculation; (**b**) the calculated roughness of the ground surface with abrasives of different sizes.

**Figure 5 micromachines-16-01049-f005:**
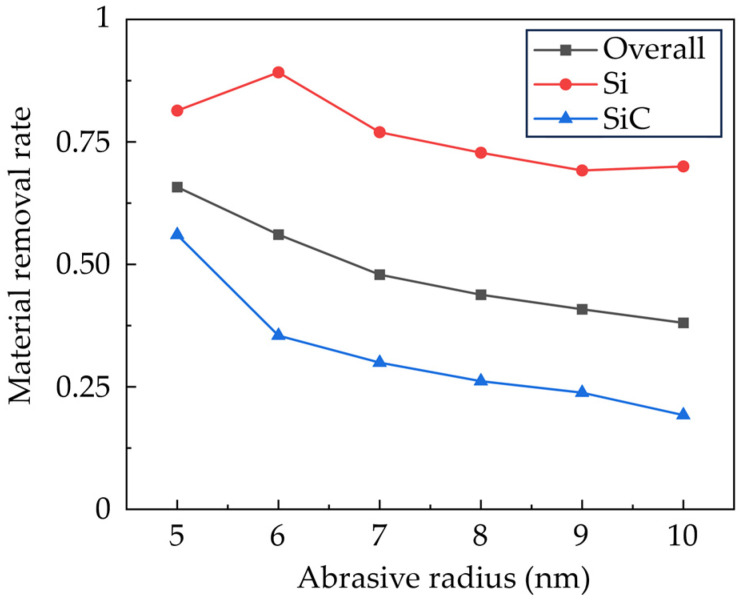
The material removal rate of the workpiece machined by abrasives of different sizes.

**Figure 6 micromachines-16-01049-f006:**
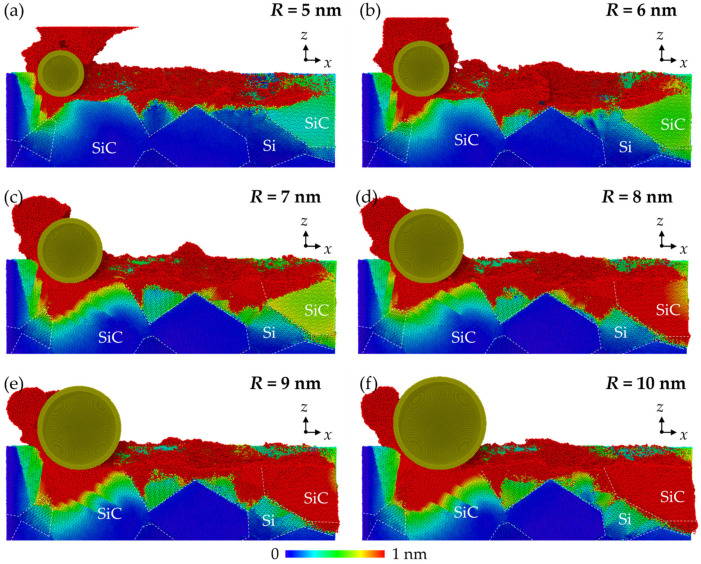
Cross-sectional views of atomic displacement magnitude distribution in workpieces machined by abrasives with radii of (**a**) 5 nm, (**b**) 6 nm, (**c**) 7 nm, (**d**) 8 nm, (**e**) 9 nm, and (**f**) 10 nm.

**Figure 7 micromachines-16-01049-f007:**
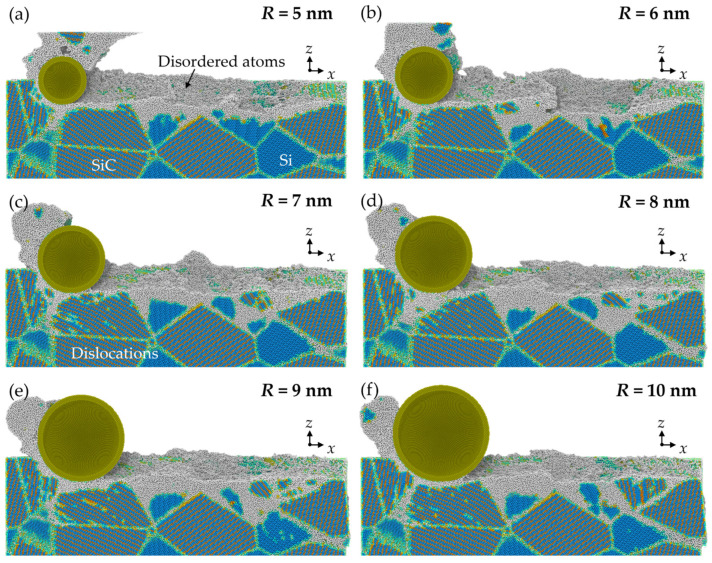
Cross-sectional view of the crystal structure in workpieces machined by abrasives with radii of (**a**) 5 nm, (**b**) 6 nm, (**c**) 7 nm, (**d**) 8 nm, (**e**) 9 nm, and (**f**) 10 nm.

**Figure 8 micromachines-16-01049-f008:**
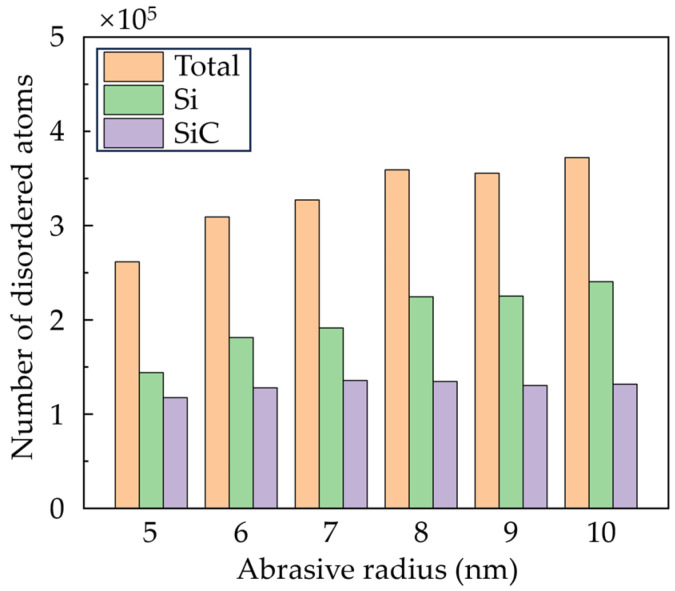
Number of disordered atoms in a workpiece machined by abrasives of different sizes.

**Figure 9 micromachines-16-01049-f009:**
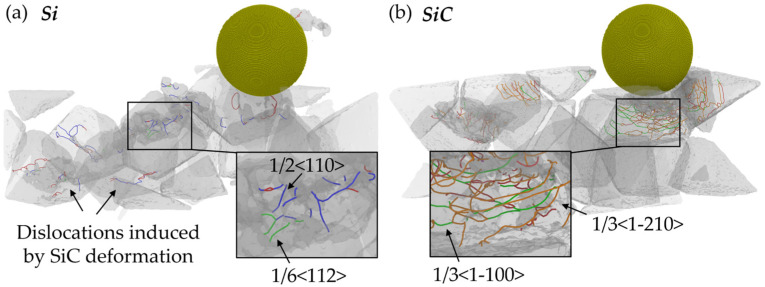
Snapshots of the dislocations in the (**a**) Si and (**b**) SiC grains of the workpiece machined by an abrasive with a 10 nm radius.

**Figure 10 micromachines-16-01049-f010:**
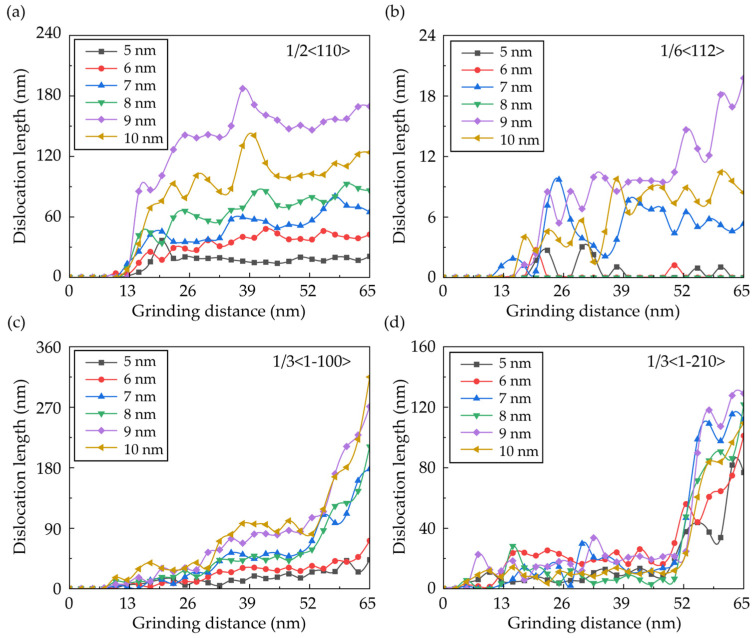
Variation of the dislocation length in a workpiece machined by abrasives with different radii: (**a**) 1/2<110> dislocations in the Si phase; (**b**) 1/6<112> dislocations in the Si phase; (**c**) 1/3<1-100> dislocations in the SiC phase; and (**d**) 1/3<1-210> dislocations in the SiC phase.

**Figure 11 micromachines-16-01049-f011:**
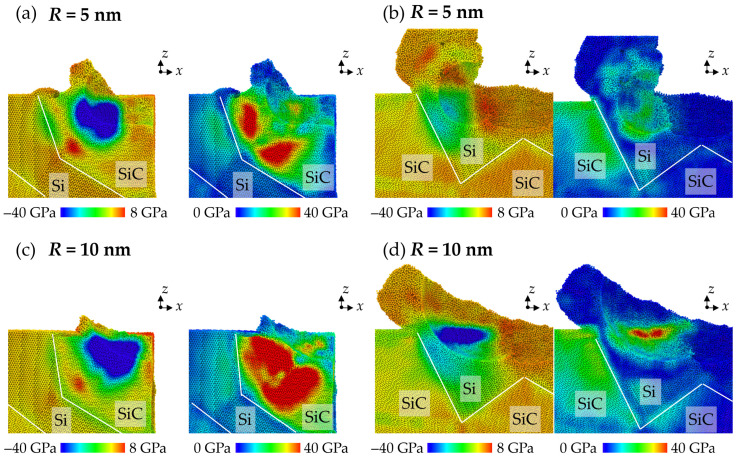
The stress distribution when machining across the phase boundaries with abrasives with radii of (**a**,**b**) R = 5 nm and (**c**,**d**) R = 10 nm.

**Figure 12 micromachines-16-01049-f012:**
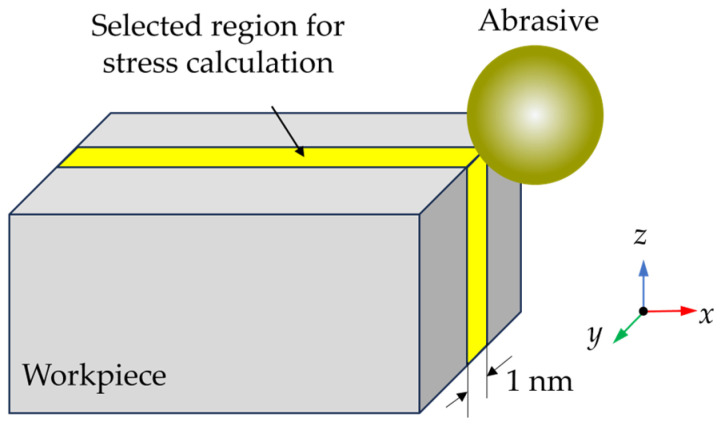
Illustration of the region selected for stress calculation.

**Figure 13 micromachines-16-01049-f013:**
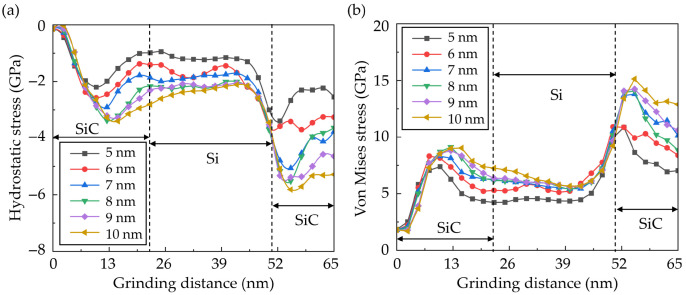
The average (**a**) hydrostatic stress and (**b**) von Mises stress of the selected region during nano-grinding with abrasives of different sizes.

**Table 1 micromachines-16-01049-t001:** Parameters of ABOP.

	Si	C
*D*_0_ (eV)	3.24	6.00
*r*_0_ (Å)	2.232	1.4276
*S*	1.842	2.167
*β* (Å^−1^)	1.4761	2.0099
*R* (Å)	2.82	2.00
*D* (Å)	0.14	0.15
*γ*	0.114354	0.11233
*c*	2.00494	181.910
*d*	0.81472	6.28433
*h*	0.259	0.5556
2*μ* (Å^−1^)	0.0	0.0

**Table 2 micromachines-16-01049-t002:** Details of the simulation parameters.

Parameters	Value
Workpiece dimension (*x* × *y* × *z*)	70 nm × 40 nm × 25 nm
Number of workpiece atoms	About 5 million
Abrasive radius (*R*)	5 nm, 6 nm, 7 nm, 8 nm, 9 nm, 10 nm
Grinding speed	50 m/s
Grinding depth	5 nm
Grinding distance	65 nm
Grinding temperature	300 K
Number of grains in workpiece	20 (10 Si grains and 10 SiC grains)

## Data Availability

The original contributions presented in this study are included in the article. Further inquiries can be directed to the corresponding author.

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
