# Peer review of "Atomic Insight into the Nano-Grinding Mechanism of Reaction-Bonded Silicon Carbide: Effect of Abrasive Size"

_micromachines, 2025, doi:10.3390/mi16091049_

Round 1

Reviewer 1 Report

Comments and Suggestions for Authors

This paper presents a molecular dynamics (MD) simulation of nano-scratching for studying the effect of abrasive grain size on the surface formation and subsurface damage of RB-SiC. The findings offer valuable insights and introduce several novel perspectives derived from the simulations. Nevertheless, certain aspects of the manuscript require further refinement.

  1. Why the author chose ABOP potential instead of the widely used Tersoff potential?
  2. The presentation of force-related results appears incomplete; could the authors provide data on the variation of cutting forces during the simulation?
  3. The machining speed employed in the simulation (50 m/s) significantly exceeds typical experimental values. How might this choice influence the simulation results?
  4. Does the distribution and orientation of SiC grains affect the simulation results?
  5. Did the author use time or spatial average on the stress tensors since the values calculated from LAMMPS could fluctuate and influence stress results.
  6. The language can be further polished.

Author Response

  1. Why the author chose ABOP instead of the widely used Tersoff potential.

R1: Thanks for the comments. ABOP is proposed in 2005 by Erhart and Albe. This potential has been successfully used to describe the interactions for several covalent materials, including SiC and Si. Compared to Tersoff potential, ABOP process much higher computational efficiency with desirable descriptions of the deformation process that has been verified by experiments [1-3].

Reference:

[1] Liu, C.; Ke, J.; Yin, T.; Yip, W.S.; Zhang, J.; To, S.; Xu, J. Cutting mechanism of reaction-bonded silicon carbide in laser-assisted ultra-precision machining. International Journal of Machine Tools and Manufacture 2024, 203, 104219,

[2] Zhen Li, Yifan Li, Liangchi Zhang, On the deformation mechanism and dislocations evolution in monocrystalline silicon under ramp nanoscratching, Tribology International 193 (2024) 109395

[3] Liu, C.; Chen, X.; Ke, J.; She, Z.; Zhang, J.; Xiao, J.; Xu, J. Numerical investigation on subsurface damage in nanometric cutting of single-crystal silicon at elevated temperatures. Journal of Manufacturing Processes 2021, 68, 1060-1071,

  1. I didn’t see the results of forces, how about the variation of cutting forces.

R2: Thanks for the comments. Machining forces are important factors for monitoring the material removal process. While in this paper, we didn’t calculate the machining forces since obvious fluctuation of the forces can be generated due to the property mismatch between phases, which has been revealed by previous research. Besides, the material removal process is investigated by analyzing the motion of workpiece atoms in a more direct view than monitoring the machining forces.

  1. The author used 50m/s as the machining speed, which is much larger than experiments. How about the influence of the machining speed on simulation results?

R3: Thanks for your comments. Due to the constraint of computational cost, the machining speed in MD simulation (range from 50m/s to 500m/s) is usually much larger than experiments to save the simulation steps. High-speed machining could cause undesirable temperature raise and high-strain rate deformation, which influence the material removal process. Therefore, a reliable MD simulation should decrease the machining speed as much as possible to minimize the high-speed effect. According to previous research, the machining speed of 50m/s has a good computational efficiency with reliable results which have been verified by experiments [1,2].

Reference:

[1] Hongfei Tao, Qinyang Zeng, Yuanhang Liu, Dewen Zhao, Xinchun Lu, Influence ofanisotropy on material removal and deformation mechanism based on nanoscratch tests ofmonocrystal silicon, Tribology International 187 (2023) 108736.

[2] Haoxiang Wang, Zhigang Dong, Renke Kang, Shang Gao, Surface characteristics and material removal mechanisms during nanogrinding on C-face and Si-face of 4H-SiC crystals: Experimental and molecular dynamics insights, Applied Surface Science 665 (2024) 160293.

  1. Does the distribution and orientation of SiC grains affect the simulation results?

R4: Thanks for the comments. The distribution and orientation of SiC grains does affect the simulation results since in ultra-precision machining, grain boundaries and crystal orientations are important factors which determine the machining mechanism. Hence in this simulation, to study the effect of abrasive size, we adopted the same workpiece in all the simulation groups to eliminate the influence of grain boundaries and crystal orientations on machining mechanism.

  1. Did the author use time or spatial average on the stress tensors since the values calculated from LAMMPS could fluctuate and influence stress results.

R5: Thanks for your comments. In this simulation, we used a spatial average of a cubic box of 2nm to average the stress tensor. In the revised manuscript, we added statement of the spatial average. (Lines 246-247)

  1. The language can be further polished and improved.

R6: Thanks for your comments. In the revised manuscript, we improved the language of this manuscript with the assistance of a native speaker.

Reviewer 2 Report

Comments and Suggestions for Authors

In table 2 please include information on assumed hardness of machined workpiece used in simulation as well as used abrasive grain.

Please indicate the size of SiC grains used in simulations.

Please discuss possibility of extrapolating presented results to the range of micrometers (when abrasive grain diameter is 1000 bigger than in presented study - it will be more interesting for potetnial readers. 

EDITORIAL CORRECTIONS:

  • line 27: shall be (RB-SiC);
  • line 61: please insert explanation for GBs shortcut (move it from line 67);
  •  

Author Response

  1. In table 2 please include information on assumed hardness of machined workpiece used in simulation as well as used abrasive grain.

R1: Thanks for your comments. In this simulation, the size of the material removal thickness is comparable to the grain size, so the hardness of the machined workpiece can be different in Si and SiC phase. Besides, the diamond abrasive is regarded as a rigid body and its deformation is not investigated in this simulation. To give a clear description about the ABOP described material, we added references about the calculated mechanical properties of SiC and Si structure in the revised manuscript. (Lines 98-101)

  1. Please indicate the size of SiC grains used in simulations.

R2: Thanks for your comments. We added relative descriptions in the revised manuscript. (Lines 94-95)

  1. Please discuss possibility of extrapolating presented results to the range of micrometers (when abrasive grain diameter is 1000 bigger than in presented study - it will be more interesting for potential readers. 

R3: Thanks for your comments. In this simulation, we investigated the effect of abrasive size on nano-grinding mechanism of RB-SiC at nanoscale. Material deformation at the interfaces is discussed with consideration of the property mismatch between phases. When the grain size increase to micrometers, the machining process when the abrasive cut into the grains is similar to the single crystals. While the result in this research is still suitable when the abrasive cut through the phase boundaries. As commented by the reviewer, we added some discussion in the revised manuscript. (Lines 83-89;182-184; 281-284)

3. EDITORIAL CORRECTIONS:

line 27: shall be (RB-SiC);

line 61: please insert explanation for GBs shortcut (move it from line 67);

R3: Thanks for your comments. We are sorry for our carelessness in presentation. In the revised manuscript, we checked the entire paper and corrected these errors.